# A Novel Simplified Protocol for Pre-Processing Whole Wood Samples for Stable Isotope Analysis in Tree Rings

Osvaldo Pericolo [1,*], Camilla Avanzi [2], Andrea Piotti [2], Francesco Ripullone [3] and Paola Nola [1]

1   Department of Earth and Environmental Sciences, University of Pavia, Via S. Epifanio 14, I-27100 Pavia, Italy
2   Institute of Biosciences and BioResources (IBBR), National Research Council (CNR), Via Madonna del Piano 10, Sesto Fiorentino, I-50019 Florence, Italy
3   School of Agricultural, Forestry and Environmental Sciences, University of Basilicata, Viale dell'Ateneo Lucano 10, I-85100 Potenza, Italy
*   Correspondence: osvaldo.pericolo@unipv.it

**Abstract:** In the context of climate change, the stable isotope analysis of tree rings may play a crucial role in deciphering the eco-physiological mechanisms underlying forest decline and dieback phenomena. However, this technique is often considered expensive, time-consuming, and with several methodological constraints. Specifically, milling and transferring the material from jars to vials during the different steps of sample preparation involve risk of contamination among samples and loss of sample material. When dealing with declining trees (i.e., trees affected by loss of vitality with strong percentage of defoliation and reduction in growth) and trees subjected to extreme events or negative pointer years (characterized by extremely narrow ring) the sample preparation is particularly difficult because of scarce amount of wood material. In such a case, pooling rings from several years to achieve the minimum weight of wood is often necessary, thus losing information at the annual resolution. In order to overcome such limitations, we developed a novel protocol for quick and accurate whole-wood pre-processing, testing it on oak tree rings of different widths taken from living trees. The main novelty introduced by our protocol was freezing tree-ring samples at −80 °C and milling multiple samples at a time by using a 24-tube plate. The results showed that our novel simplified protocol significantly reduced the pre-processing time with respect to the standard protocol (12 vs. 284 sec/sample), while achieving the same wood particle size, limiting the loss of wood material and reducing the risk of contamination among samples.

**Keywords:** stable isotopes wood pre-processing; lab work optimization; water-use efficiency; climate change; forest decline

## 1. Introduction

Increasing rates of forest decline and die-off events have been reported worldwide [1,2]. These trends have been attributed to direct and indirect impacts of water stress and rising temperatures, especially in drought-prone environments, such as that of the Mediterranean basin [3–5]. The consequences of this phenomenon have a large impact on both short-term forest functioning [6] and long-term ecosystem dynamics [7,8]; however, our understanding of the physiological mechanisms leading to forest decline is still limited. In this context, the analysis of the isotopic signals within tree rings could play a crucial role to obtain retrospective information concerning the eco-physiological responses of declining trees [9,10]. In particular, the reduced stomatal conductance and the limited photosynthetic rate imposed by water stress trigger variations of isotope fractionation during both $CO_2$ uptake and fixation in the leaves and, subsequently, in the wood.

Despite the valuable information provided by isotopic analysis, this practice is still severely limited by costs, time, and very often by the exiguous amount of wood in each individual tree ring [11]. Properly processing tree ring samples involves a lengthy preparation phase, which has several methodological constraints, such as the risk of contamination

among samples belonging to different years or trees, the loss of material during the milling and the transfer in the vial, and the achievement of optimal particle size to ensure homogeneity in the sample. Overall, such limitations often impede extending the analysis to a large number of individuals or sites. In addition, isotopic analysis has frequently been limited to a few years [12–14], often preventing inference on the long-term inter-annual climate variability and the related physiological processes of tree growth. The pre-processing of whole wood samples is particularly challenging when dealing with extremely narrow tree rings (<1 mm), as in declining trees (i.e., trees with crown defoliation $\geq$ 50% and a strong reduction in radial growth, [15]) or after extreme events, whose data are key for understanding eco-physiological processes during stressful episodes. When tree rings are so small (up to 0.12 mm and to 0.0008 g), it is often necessary to pool rings from several years to achieve the minimum quantity of wood necessary for isotopic analysis (up to ten tree rings [10,16]), losing information at the annual resolution [17].

In order to overcome such limitations, we search for a novel procedure to speed up milling tree-ring samples, reduce the risk of contamination among samples, and limit the loss of wood material during laboratory steps. Notably, developing a new simplified protocol that enables researchers to process more tree ring samples per time could enlarge the investigation range based on isotopic analyses.

## 2. Materials and Methods

### 2.1. Sample Material and Pre-Treatment

The wood material used in the present experiment came from tree-ring incremental cores extracted with a Pressler borer at breast height from declining and non-declining English oak (*Quercus robur* L.) trees. We selected this species because oak produces hardwood that is more difficult to work (cut and grind) than softwood. In addition, dealing with declining and non-declining trees gave us the opportunity to include extremely narrow and large tree rings.

Increment cores were air-dried and polished with a scalpel until the lumens of the vessels were clearly visible. After that, single tree rings were separated from each other manually using a scalpel under a stereoscope. After cutting, each tree ring was placed in a 2.0 mL Eppendorf tube labeled with the ID of the wood sample.

In order to test the efficiency of milling on tree-ring samples of different sizes, we clustered tree rings into four size classes (hereafter referred to as classes I, II, III, and IV), depending on their width. Specifically, class I included samples with tree-ring width < 1.0 mm, class II included samples between 1.0 and <2.0 mm, class III between 2.0 and <4.5 mm, and IV $\geq$ 4.5 mm. Samples included in classes III and IV were further divided into smaller pieces to make the next grinding step easier. This reduced the grinding time in the case of large samples (classes III and IV) and reduced the probability to damage the Eppendorf tubes, thus decreasing the risk of introducing microplastics into the wood sample.

Before milling, all samples were dried in a laboratory oven at 50 °C for four hours. After this time, additional drying did not increase water loss from samples.

Five samples for each class were then processed by applying two different protocols.

### 2.2. Standard Pre-Processing Protocol

For the standard milling procedure, tree-ring samples were ground using a mixer mill (MM 400, Retsch GmbH, Haan, Germany, Figure 1A) equipped with two grinding stations. The material of each annual ring was milled individually to a fine powder in dry mode inside a grinding jar, coupled with a stainless-steel ball 25 mm in diameter (Figure 1C). The milling duration was 20 s (one cycle), with a frequency of 20 Hz.

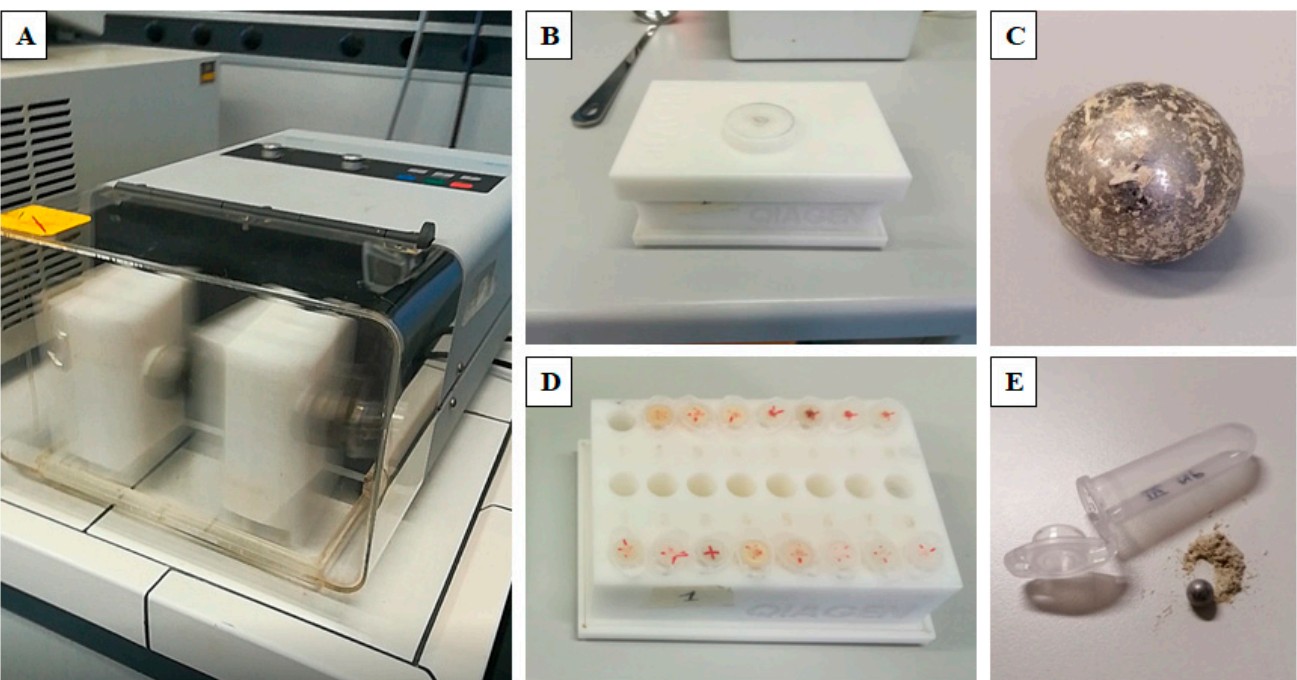

**Figure 1.** Equipment for milling tree-ring samples with standard and new protocol. Panel (**A**) shows the Mixer mill MM 400 (Retsch GmbH, Haan, Germany) used for milling tree rings in both protocols. Panels (**B**,**D**) show the plate used for multiple milling of tree-ring samples (up to 12 samples in each plate). Panel (**C**) shows an example of loss of wood material with the standard protocol, in particular, the milled material attached on the grinding ball which is particularly hard to retrieve. Panel (**E**) shows the small steel bead used and the wood powder obtained after milling.

After milling, the obtained wood material was collected from the grinding jars and transferred into the Eppendorf tubes with great care. Then, the two grinding jars and the two steel beads were cleaned and washed for reuse with ethanol to remove any wood residual. The duration of each step was timed and recorded. We refer to this procedure as "standard protocol" for the sake of simplicity. We are aware that different laboratories adopt several variants of such a procedure; our aim here was to compare the new pre-processing protocol with a frequently used ball-milling protocol described in the literature.

*2.3. New Pre-Processing Protocol*

In the new protocol, after the drying step, a stainless-steel bead 5 mm in diameter (Qiagen, Germany) was added to each Eppendorf tube (with safety lock caps) containing tree-ring samples. Then, the Eppendorf tubes were stored in a −80 °C freezer for at least 12 h before milling. When the material was completely frozen, they were inserted into a 24-tube plate (Qiagen, Germany), which permitted simultaneous milling. The low temperature, coupled with the movement of the stainless-steel bead inside the Eppendorf tube, allowed the breaking and the milling of tree-ring samples.

To ensure effective milling of the wood samples, some steps were crucial. Firstly, it was important to insert the steel bead before storing the samples in the freezer so that the bead reached the same temperature of the sample. Quickly milling the samples was required to minimize warming after removal from the freezer. If the time at room temperature is prolonged for any reason, it was important to refreeze the samples before proceeding with the additional grinding.

We used the same mixer mill employed in the standard protocol by fixing two plates to grinding stations (Figure 1A,B,D). The grinding plate of the mixer mill performed radial oscillations in a horizontal position with a vibrational frequency equal to 30 Hz. The inertia of the grinding beads caused them to impact with high energy on the frozen sample inside

the Eppendorf tube and pulverize it. Additionally, the combined movement of the plate and the beads resulted in the intensive milling of the sample (Figure 1E).

After some tests, we fixed a standardized operating procedure, described as follows. First, as we noted that samples placed on the middle line of the plate did not mill well, we decided to use only the two external lines to improve the milling performance (placing empty Eppendorf tube in the 4 corners to balance the plate during the grinding step), thus we processed a maximum of 12 samples per plate, totaling 24 samples per run. We fixed the grinding time at 60 s, divided in two cycles of 30 s each. Between the two cycles, the plates were rotated 180° from the initial position to ensure a homogeneous milling between the two selected plate lines. The duration of all steps was recorded. The presence of non-wood objects, such as microplastics, in the milled material was initially checked by an optical stereo microscope (Leica, Wetzlar, Germany). In addition, we tested the likely presence of microplastic particles, not detectable by the optical approach, with Raman spectroscopy, using LabRAM HR Evolution (HORIBA UK Ltd., Stanmore, UK). In particular, we compared the polypropylene spectrum obtained from the Eppendorf tube and the spectra from ten samples milled with standard and new protocols. For each measure, we made 10 repetitions, randomly distributed within the sample.

Panel A shows the Mixer mill MM 400 (Retsch GmbH, Haan, Germany) used for milling tree rings in both protocols. Panels B and D show the plate used for multiple milling of tree-ring samples (up to 12 samples in each plate). Panel C shows an example of the loss of the wood material with the standard protocol, in particular, the milled material attached on the grinding ball that was particularly hard to retrieve. Panel E shows the small steel bead used and the wood powder obtained after milling.

### 2.4. Analysis of Wood Particle Size from Different Protocols

To evaluate the efficiency of this new protocol, we compared the results of sample milling between the two protocols through a physical characterization of wood particles. Specifically, the particle-size distribution of the two milling protocols was measured using a laser diffraction particle size analyzer (Mastersizer 3000, Malvern Instruments Ltd., Worcestershire, UK) coupled with a Hydro EV wet powder dispersion accessory. The particle-size distribution was calculated from the light scattering pattern based on Fraunhofer theory, using six replicated measures for five tree rings for each of the four size classes and each of the two protocols ($6 \times 5 \times 4 \times 2 = 240$ distributions). The particle-size distributions (PSDs), i.e., particle size at 1% (Dv1, minimum diameter), 50% (Dv50, median diameter), and 100% (Dv100, maximum diameter) of the volume distribution, were all calculated automatically using a specific SOP (Standard Operating Procedure) set for wood particle analyses. To avoid the flotation of larger wood particles within the water, we used a 1:1 solution of water and denatured alcohol.

### 2.5. Statistical Analyses

Linear intercept mixed-effect models were used to assess whether the two applied protocols led to significant differences in the size of wood particles. The analyses were performed using the "nlme" and "car" packages [18,19] of the R statistical suite [20]. Mean, median, minimum, and maximum particle sizes were used as response variables. Two categorical variables, i.e., protocol and tree-ring size class, were included in the model as fixed effects, together with their interaction. The rationale beyond each model was to check that there was no difference in the size of wood particles between the two protocols for each tree-ring size class. Thus, a non-significant effect was expected for both "protocol" and "protocol × class" variables. A RingID variable was included in the model as a random factor to account for the variance due to the different rings analyzed for each size class. The within-group variance structure was also taken into account by using the varFunc constructor available for the "lme" R function. For each response variable, four different models with a different combination of the variance structure of the two fixed effects were tested, choosing the best model as the one with the lowest AIC value.

### 3. Results

The time length of each step of the two protocols is summarized and compared in Table 1. Given that the standard protocol worked with two samples at a time, while the new protocol with 24 samples, for comparison purposes, Table 1 summarizes both the total time for each step and the time related to process a single sample. Freezing time was not considered because it was not counted as manhours and not linked to the number of samples to be processed. For lab work optimization, this time only needed to be taken into account when scheduling the milling sessions.

**Table 1.** Timing comparison between the standard and the new protocol for milling one tree-ring sample.

| Steps of the Standard Protocol | Total Time Per 1 Run (2 Samples per Time) (Seconds) | Time per 1 Sample (Seconds) | Steps of the New Protocol | Total Time per 1 Run (24 Samples per Time) (Seconds) | Time per 1 Sample (Seconds) |
|---|---|---|---|---|---|
| Adding steel balls into grinding jars | 4.0 | 2.0 | Adding steel beads into each Eppendorf tube | 48.0 | 2.0 |
| Fixing and adjusting jars in the mill | 25.0 | 12.5 | Fixing and adjusting plates in the mill | 25.0 | 1.0 |
| Samples milling in grinding jars | 20.0 | 10.0 | Samples milling in plates | 60.0 | 2.5 |
| Recovering the milled material from the jars and the steel balls | 330.0 | 165.0 | Removing steel beads | 132.0 | 5.5 |
| Material transfer from the grinding jar to the Eppendorf tube | 90.0 | 45.0 | Not required | - | - |
| Washing of the grinding jars and the steel balls for reuse | 99.0 | 49.5 | Washing of the steel beads for reuse | 24.0 | 1.0 |
| TIMING STANDARD PROTOCOL | 568.0 | 284.0 | TIMING NEW PROTOCOL | 289.0 | 12.0 |

The number of steps for each protocol varied (Table 1). Specifically, the new protocol did not include the sample transfer and instrumentation washing steps. The final outcome was that the total time required to process a single sample with the new protocol was about 23 times shorter than the time spent with the standard protocol.

The size of the wood particles obtained after milling varied in a wide range for both protocols (Figure 2). For the standard protocol, the particle size ranged from 1.32 μm to 3490 μm, and from 1.31 μm to 3500 μm for the new protocol. For both protocols, the minimum particle size values (1.32 and 1.31 μm) were found in tree-ring samples included in class I. Regarding the mean particle size, the values ranged from 103 μm to 603 μm for the standard protocol and from 103 μm to 679 μm for the new protocol. For median particle size, the values were in the range of 83 μm to 391 μm for standard protocol and from 87 μm to 460 μm for the new protocol. The largest variability was found for the maximum particle size included in class IV for both protocols, with particle sizes ranging from 399 μm to 3490 μm for the standard protocol and from 391 μm to 3500 μm for the new protocol.

A set of generalized linear mixed-effect models were applied to test any possible difference between the particle size of the milled material obtained by using the two protocols. The "protocol"factor ($\chi^2 = 2.14$, df = 1, $p = 0.14$) and "protocol $\times$ class"interaction ($\chi^2 = 0.67$, df = 3, $p = 0.88$) had no effect on the mean size of wood particles. Similarly, no effects of the "protocol"factor and the "protocol $\times$ class"interaction were observed for the minimum (protocol: $\chi^2 = 0.02$, df = 1, $p = 0.87$; protocol $\times$ class: $\chi^2 = 1.89$, df = 3, $p = 0.60$), median (protocol: $\chi^2 = 0.24$, df = 1, $p = 0.62$; protocol $\times$ class: $\chi^2 = 1.64$, df = 3, $p = 0.65$), and maximum size(protocol: $\chi^2 = 0.11$, df = 1, $p = 0.74$; protocol $\times$ class: $\chi^2 = 0.78$, df = 3, $p = 0.85$) of the volume distribution. A significant effect of tree-ring size class was observed for the mean size of wood particles ($\chi^2 = 62.99$, df = 3, $p < 0.001$), as well as for the minimum, ($\chi^2 = 36.31$, df = 3, $p < 0.001$), median ($\chi^2 = 42.27$, df = 3, $p < 0.001$), and maximum size

($\chi^2$ = 42.56, df = 3, $p$ < 0.001) of the volume distribution. All the results of the linear mixed model are summarized in Table 2.

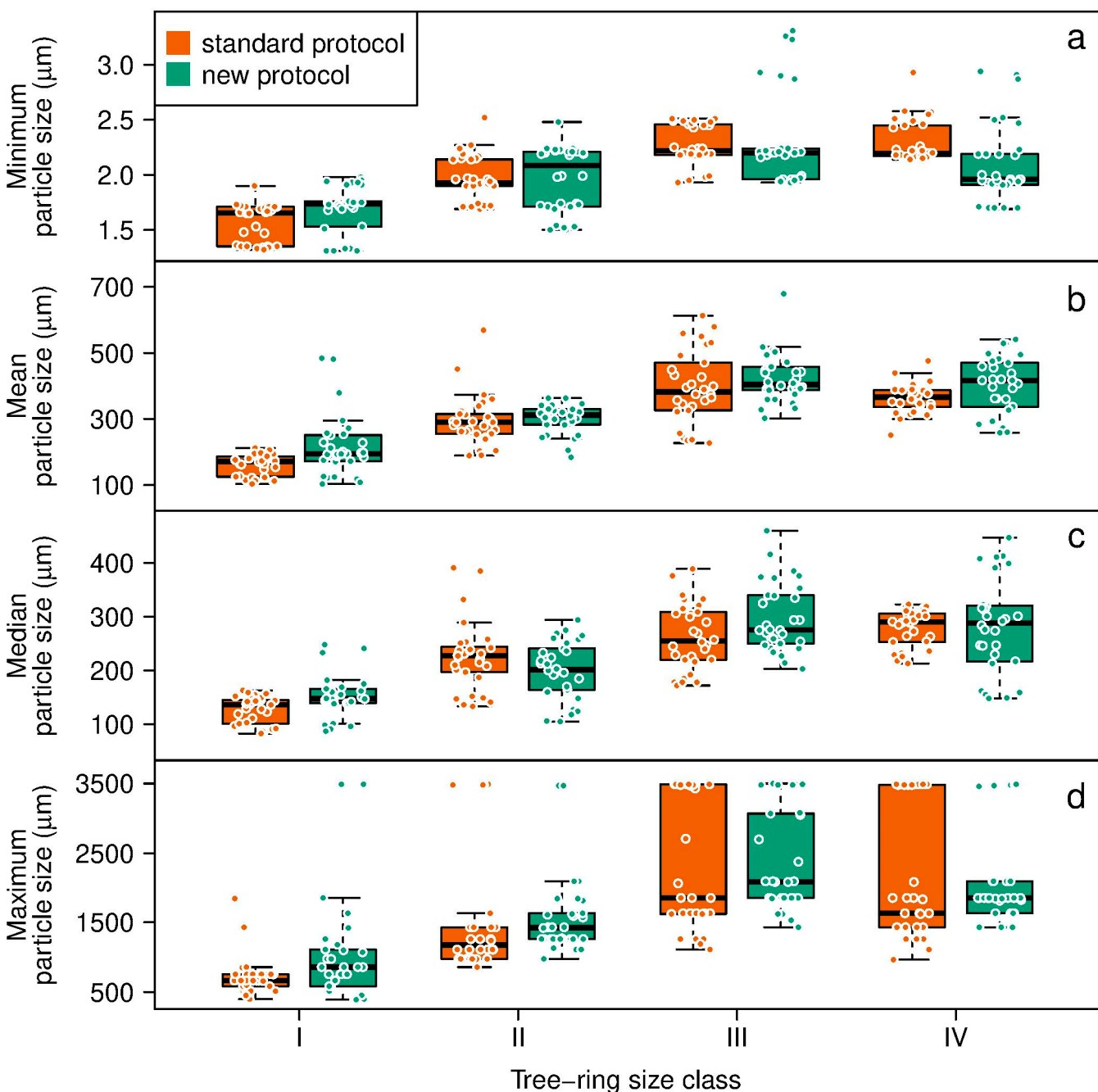

**Figure 2.** Box plots showing the minimum particle size (**a**), the mean particle size (**b**), the median particle size (**c**), and the maximum particle size (**d**) in each dimensional class of tree ring width (I, II, III, IV), comparing the standard and the new protocol. In each boxplot, data were also added by jittering coordinates along the x-axis. Measurements of particle size are in microns (μm). The range of detectable particle sizes by laser diffraction particle size analyzer is 0.01–3500 μm.

**Table 2.** Results of the Wald chi-square tests for the fixed effects (protocol, class, and their interaction), tested with generalized linear mixed-effect models. For each response variable (minimum, mean, median, and maximum wood particle size), the variance structure of the chosen model (i.e., the one with the lowest AIC value) is also reported.

| Response Variable | Fixed Effect | $\chi^2$ | Df | p ($\alpha$ = 0.05) | Variance Structure of the Chosen Model |
|---|---|---|---|---|---|
| *Minimum* | protocol | 0.241 | 1 | 0.623 | varIdent (form = ~ 1 \| class) |
| | class | 42.272 | 3 | $3.513 \times 10^{-9}$ | |
| | protocol × class | 1.639 | 3 | 0.651 | |
| *Mean* | protocol | 2.137 | 1 | 0.144 | VarComb (varIdent (form = ~ 1 \| class), varIdent (form = ~ 1 \| protocol)) |
| | class | 62.993 | 3 | $1.348 \times 10^{-13}$ | |
| | protocol × class | 0.665 | 3 | 0.881 | |
| *Median* | protocol | 0.023 | 1 | 0.879 | VarComb (varIdent (form = ~ 1 \| class), varIdent (form = ~ 1 \| protocol)) |
| | class | 36.316 | 3 | $6.420 \times 10^{-8}$ | |
| | protocol × class | 1.891 | 3 | 0.595 | |
| *Maximum* | protocol | 0.109 | 1 | 0.741 | varIdent (form = ~ 1) |
| | class | 42.564 | 3 | $3.045 \times 10^{-9}$ | |
| | protocol × class | 0.783 | 3 | 0.854 | |

After checking the absence of larger non-wood objects in the milled wood by the optical approach (microscopic examination), we tested for the presence of smaller particles, such as microplastics, by Raman spectroscopy. The Raman test did not detect any trace of microplastic as shown in Figure 3, even if we cannot completely rule out traces of non-wood objects in our milled samples. The potential impact of such untraceable contamination on isotopic data could be further assessed by analyzing the same samples prepared with the two protocols compared in our study. Certainly, the short milling time and the low frequency significantly contributed to the absence of microplastic contamination. This caution should be always carefully taken into account when preparing wood samples for isotope analysis, regardless of kind of protocol used.

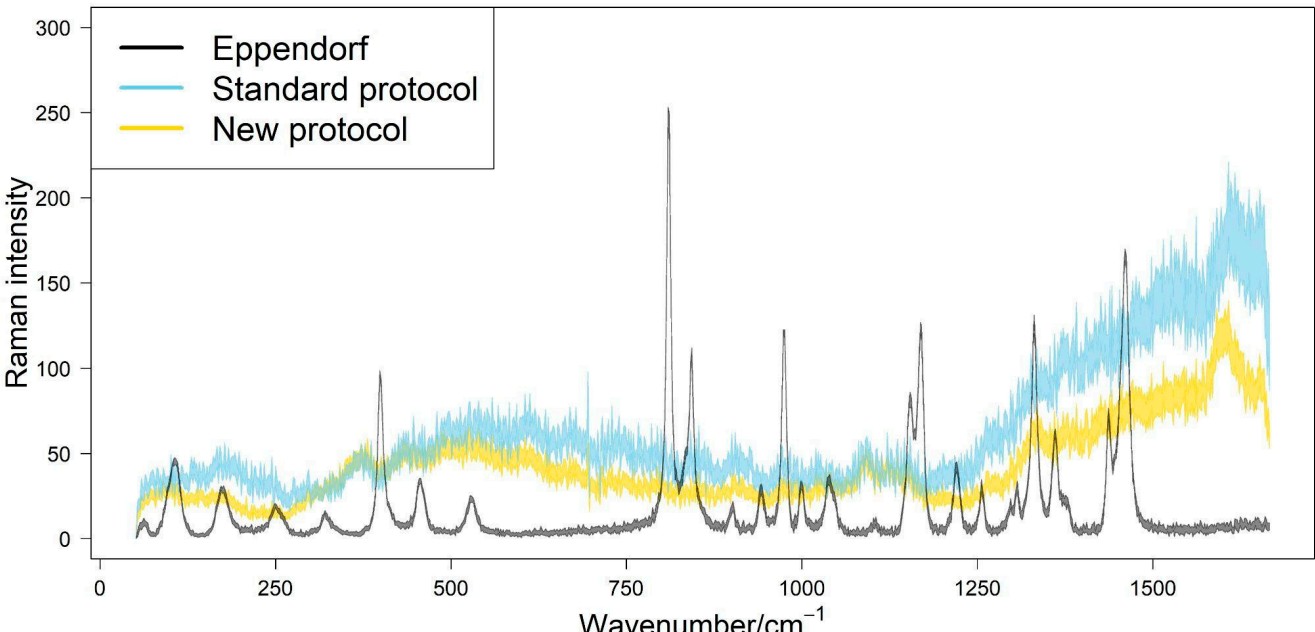

**Figure 3.** Comparison among Raman spectra of Eppendorf tubes (in grey) and wood milled with the two protocols (in colors). Each curve represents the confidence interval around the mean Raman spectra of Eppendorf tubes (in grey) and wood milled with the standard and new protocol (in light blue and yellow, respectively), over multiple Raman measures.

## 4. Discussion

Despite a recently proposed efficient method to improve isotopic analysis related to cellulose isolation [21], the literature and guidelines on preparing whole wood samples for isotopic analysis are still limited [17] and, to our knowledge, no study has ever attempted to quantitatively compare different procedures and methods. Here, we proposed a novel protocol for sample pre-processing, showing that it significantly reduced the milling time, the risk of contamination among samples, and the loss of wood material, while achieving the same particle size obtained with standard procedures. In particular, the new protocol overcame some of the disadvantages of the standard procedures, i.e., recovering the milled material from the grinding jars and steel balls, transferring the wood material from the grinding jars into an Eppendorf tube for storage, and washing the grinding jars and steel balls for reuse. These steps generally pose a risk for sample contamination and loss of wood material (Figure 1B,C), even if the risk of contamination due to cleaning the stainless-steel beads is then present also for the new protocol. The loss of material, in particular, may prevent the achievement of the minimum weight threshold required by the isotopic analysis, especially when the amount of wood material available is extremely scarce, which is very often the case for tree rings of declining trees. Instead, with our new protocol, the loss of material was negligible because the milling occurred inside the Eppendorf tube. The possibility to efficiently work when the material is scarce would allow us to study the response of declining trees with annual resolution, achieving the minimum weight threshold required for year-by-year analysis. This could be of fundamental importance in retrospectively reconstructing water-use strategies in declining and non-declining trees with respect to climate change, especially in warm and drought-prone environments, such as the Mediterranean Basin [22,23]. Secondly, but no less importantly, limiting the loss of material could ensure that the analyzed powder comprehensively represents the entire growth ring and, therefore, the full growing season will be better represented [24,25]. This is crucial for analyzing the Mediterranean tree species whose cambial activity may be interrupted by a temporary summer dormancy [26,27]. Lastly, the new protocol presented and tested here could allow the total amount of man hours to be reduced, thus encouraging the increase in the number of samples processed, even if the cost of the analysis itself remains high. Since different studies reported large variabilities among trees in parameters from isotopic analyses [28,29] and little evidence that isolating a single chemical component (e.g., cellulose) significantly reduces this variability [30,31], we deem that increasing the number of individuals analyzed and the temporal resolution of stable isotope analyses of tree rings may help to better define the population variance. Although, in an environmentally homogeneous site, usually, four-to-six trees are considered enough to establish common responses to stressful events, and the number of trees required to obtain more robust results may increase depending on individual tree features, tree species, and site conditions [17].

## 5. Conclusions

Here, we presented a new protocol for tree ring preparation to be used for stable isotope analysis that sped up the time in milling tree-ring samples, reduced the risk of contamination among samples, and considerably reduced the loss of wood material. This combination of features was important when dealing with scarce wood material (e.g., in narrow tree rings associated with extreme events) and when processing a large number of samples (e.g., to study tree responses with an annual resolution guaranteeing a comprehensive coverage of trees' lifespan), but it became essential when a large number of samples characterized by very narrow tree rings had to be analyzed to investigate the drivers of complex processes such as forest decline and dieback phenomena.

Increasing both the number of individuals/populations and the temporal resolution of stable isotope analyses of tree rings are key to advance our understanding of trees' ecophysiological responses to climate change [11]. If the impact of such advancements on eco-physiological research is quite straightforward, it also generates considerable expectations in evolutionary research. In particular, large-scale projects based on dendrogenomic

approaches (e.g., [32,33]), as well as genotype-dendrophenotype association studies both in common gardens (e.g., [34,35]) and natural populations [36–39], would greatly benefit from adding water use efficiency measurements to the battery of dendrophenotypic traits investigated. In this regard, it is important to finally stress that this new protocol has no contraindications in being used on wood of other tree species, while this technical note is focused on oak tree rings. It might, however, be adjusted depending on the intrinsic characteristics of the wood (e.g., how hard the wood is or if it contains any essential oils included in the samples) by fine-tuning the milling time and the frequency.

**Author Contributions:** Conceptualization, P.N. and O.P.; methodology, P.N., A.P., O.P. and C.A.; investigation, O.P. and C.A.; data curation, C.A., A.P., O.P. and P.N.; original draft preparation, O.P. and P.N.; review and editing, O.P., C.A., A.P., F.R. and P.N.; funding acquisition, P.N. and F.R. All authors have read and agreed to the published version of the manuscript.

**Funding:** This research was carried out in the framework of the project ResQ "Oak dieback in lowland forests: a multidisciplinary study for the selection of resistant genetic resources" (CUP F84I18000490003) co-funded by RegioneLombardia (Northern Italy), Bando 2018 d.d.s. n. 4403 del 28 March 2018. This study was also carried out within the Agritech National Research Center (CUP C33C22000250001) and partially financed by the European Union Next-GenerationEU (Piano Nazionale di Ripresa e Resilienza (PNRR)—Missione 4 Componente 2, Investimento 1.4—D.D. 1032 17 June 2022, CN00000022).

**Data Availability Statement:** All the research data used in the present work are freely available online or obtainable directly from the authors upon request.

**Acknowledgments:** We thank Francesca Bagnoli, Ilaria Spanu, and Tatiana Storchi for help in lab activities. We thank Luca Moretti for helpful suggestions for analyzing Raman spectra. We thank Manuel La Licata for the support during the wood particles characterization with the Mastersizer 3000 and Mattia Gilio for the valuable contribution with Raman Spectroscopy.

**Conflicts of Interest:** The authors declare no conflict of interest.

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
