# Peer review of "A Novel Simplified Protocol for Pre-Processing Whole Wood Samples for Stable Isotope Analysis in Tree Rings"

_forests, doi:10.3390/f14030631_

Round 1

Reviewer 1 Report

Pericolo et al. compare two pre-processing protocols for milling tree ring samples, relevant for whole wood isotope analysis. It is, however, not clear from beginning, that they aim to present a pre-processing for whole wood samples only - for cellulose samples, other protocols are nowadays established. Thus, the new protocol might not be useful for cellulose samples.

I also wonder why the authors define delta values (C only - why nothing about O/H?) in the introduction, while there is nothing about isotope analysis in the remaining manuscript. Regarding this, I, however, think that it is important to check for isotopic changes due to the adjusted pre-processing. I am concerned about milling in Eppendorf tubes using small steel beads - how to be sure that no tiny pieces of tube material remain in the sample (microplastics), thus, having an effect on the isotope results?! This needs to be checked. Single tree rings have to be cut into several pieces and split into two groups, one milled with the old protocol, the other with the new protocol.

With respect to the protocol changes, I also wonder, why not comparing the effect of freezed samples directly milled in the mill jars (instead of Eppendorf tubes). This clearly would have an effect. In the present case, the authors compare non-freezed milling in mill jars with milling freezed samples in Eppendorf tubes. Freezing the samples already has some effect and might also be beneficial when milling the samples in the beakers. Currently, the overall effect of the new protocol can also be due to freezing - you, however, do not compare this. Freezing the samples is not new.

Detailed comments:

line 2f: you should mention that this protocol is for whole wood analysis only (or if not: clarify this in the manuscript and discuss how your pre-processing would be applicable for cellulose isolation where it is more common to use filter bags for analyses)

l. 11ff: as above: you have to mention that this is about whole wood pre-processing (otherwise clarify; see above); moreover, after reading the abstract I expected to read something about isotopes in this manuscript

l. 12: “deciphering eco-physiologcial mechanisms underlying forest decline” - this is not part of this study but the reader might think that your manuscript addresses this topic

l. 15: maybe use “sample material” instead of “wood”

l. 18: how to define “narrow” (in mm)?

l. 19: “resolution [level]”

l. 20: clarify what is new in your protocol, already in the abstract - and be accurate: it is about pre-processing/milling, not about any extraction procedure (whole wood)

l. 21: “oak tree rings” - talking about modern trees, I guess?

l. 25 “Tree ring milling” is a somehow strange keyword, I would say

l. 25f “Phenotypic characterization” - what is this and what does this have to do with this manuscript?

l. 49 After defining the delta value, I was sure to read something about isotopes in the manuscript. Moreover, why are you only writing about C, not about O or H? As written above, it seems necessary to me to include isotope analysis to check the impact of the changed protocol (see also below for details)

l. 61 “contamination” - please give more details here

l. 64f “limited to a few years” - why?

l. 67 “extremely narrow” - what is narrow? what is extremely narrow? (in mm)

l. 69 “so small” - how small?

l. 71 “resolution [level]”

l. 83 ”declining and non-declining” - how can this be selected?

l. 95f I am a bit unsure about the exact borders: class II from 1.0 to <2.0 or ≤2.0?

l. 96 “further divided into smaller pieces…” please give more details. Is there one tree ring within a single Eppendorf tube or splitted into several ones (in case of the larger classes)?

l. 101 only talking about pre-processing?

l. 102f Please give references for the “standard protocol”

l. 105 “…steel bead of 300 mm in diameter” - you could refer to Fig. 1C here

l. 107 “wood material was collected from the grinding jars” - how? Did you try to remove the powder from the jars and the bead (this usually can quite easily be done using e.g. spatula and something like weighing paper)

l. 108 “vials” - you mean Eppendorf tubes?

l. 109 “cleaned and washed for reuse” - how? Do you include a ethanol run in the mill? This is very efficient

l. 113 “stainless steel bead” There are information required on the manufacturer, maybe also include an image within Fig. 1?

l. 114f “..samples. Then, …”

l. 115 “-80°C freezer” - it seems mandatory to compare milling freezed samples for both milling procedures. Why were the samples in the standard milling protocol not freezed?

l. 116 “24-tube plate” - give information on the manufacturer.

l. 117 “movement of the stainless steel bead inside the Eppendorf tube” - I am highly concerned about potential microplastics resulting from this procedure. Eppendorf tubes are not made for milling so you cannot be sure that no microplastics were mixed with your samples. It is mandatory to include a isotope analysis here to exclude this. Use single tree ring samples and split them into two groups. While one group is milled with the old protocol (although, I would also suggest to freeze the sample before milling), the other group is milled using the new protocol. Then check the isotope data (C/O/H!). Otherwise, think about using other vials where this can be excluded (however, this would probably increase the costs and maybe also time, if you cannot store the samples in these vials)!

l. 122 “bead reaches the same temperature of the sample” - why is this important? To prevent heating up the sample?

l. 127 “we used a special plate” - manufacturer? How do you fix this in the Retsch mill (include an image within Fig. 1 and describe this in detail)? How do you fix the Eppendorf tubes (prevent falling out)? Does the plate has a lid? I also wonder how to use the Retsch mill without the jars (I have experience with this mill)- is the weight of the plate sufficient for the mill to start milling (and not stopping because of missing jars)? Do you use two plates in parallel?

l. 132 “the beads resulted in the intensive mixing of the sample” - this is probably also the same for the standard protocol

l. 132 What should be demonstrated with Fig. 1B?

l. 141 “foreign objects” - but you won’t see microplastics!

l. 156 “reliability” - with respect to what?

l. 178 “RingID variable” - I assume this is just the specimen number? So you have ID X for both the standard and the new protocol?

l. 185 Please include details on the chi square test and mention the alpha level used in this study

l. 189 doubled spacing between “purposes,” and “time”?

l. 190 “time needed to process a single sample” - the whole time overview (Table 1) is a bit confusing and needs more explanation and maybe some rearrangement etc. of the table

Table 2: first step standard protocol - not required: well, you have to add the bead to the jar, as well

Table 2: adjusting the jars vs. adjusting the plates in the mill - time required?

Table 2: standard protocol: “recovering the milled material…: 30 s” vs. “Material transfer from the grinding jar: 265 s” - I am not sure how these two steps differ? You can simply remove the bead, emty the jar on a, e.g., weighing paper, remove the attached wood powder from the jar and the bead and fill everything in the Eppendorf tube (starting from the weighing paper). 295 s for removing material and transferring it seems quite inefficient to me.

Table 2: “washing of the grinding jars…” - as asked above: does this include a run with ethanol?

Table 2: new protocol: “check to verify the milling results” - this is obviously also required for the old protocol

Table 2 Why is washing not required for the small beads? Do you use them only once? Why? This seems to be a very expensive thing over the time

l. 215 “maximum particle size included in class IV for both protocols” - please also give details here (as for mean etc.)

Fig. 2 I am a bit unsure about the sample size. According to l. 100 you use 5 samples per class? But for, e.g., class III minimum new protocol you have more than 5 measures in the image (already 5 outliers, some others in the box). Please specify the exact numbers - already above.

Fig. 2 Please add a jitter plot to these box plots (just above the boxes)

Fig. 2 You also have to talk about the detected outliers! Are these specific samples? Are they included in your chi square tests?

l. 225 degree of freedom is more commonly given as “df” instead of “d.f.” (also in the following)

l. ….P=0.65)[,] and…”

l. 233 remove comma after “minimum”

l. 240 “variance structure of the chosen model…” This is not mentioned in the text. Please explain this in the text (and also discuss it below)

l. 245 “cellulose isolation” instead of “extraction”

l. 245f At this point one could assume that this is about whole wood only - not mentioned before

l. 247 ref. [21]: this is not a protocol for whole wood, rather cellulose, and details on the protocol are missing; use other references here

l. 247 “no study has ever attempted to quantitatively compare different procedures and methods” - talking about pre-processing only? Otherwise, regarding the whole protocol (i.e. for cellulose, not sure about whole wood) this is incorrect

l. 249 “sample preparation” - it is rather pre-processing, I would say

l. 249f In the beginning you also mentioned the costs - what about the costs? (i.e. if you really use the small beads only once)

l. 250 “risk of contamination” - but how to clean the small steel beads?

l. 253 As mentioned above, this might be also different when using freezed samples! This has to be checked as well.

l. 255 What about washing the beads of the new protocol? Loss of wood material is not a large factor here, but still there could be contamination if not sufficiently cleaned. Or are they used only once as one could assume according to Table 2 (see my comments above)?

l. 257 “minimum weight threshold” - namely? You also have to discuss why particle size is important for isotope analysis anyway. What is the effect of particle size on isotope analysis?

l. 259 “the loss of material…” Please also provide data on the weight of the unmilled vs. milled samples for both protocols and include this in the results section.

l. 272 “large savings in man hours” Is this that high, i.e. compared to mass spectrometry costs - please specify this a bit more

l. 275 “significantly reduces this variability” - what is this now about?

l. 284 “Here, ….” please rewrite this sentence

l. 296f “dendrogenomic approaches” - Is the milling protocol applicable for these analyses?

l. 301 “this new protocol has no contraindications in being used on wood of other tree species” - It might, however, be adjusted depending on, e.g., how hard the wood is or maybe also dependent on the amount of essential oils included in the sample (e.g., adjusted milling time). Thus, this should be included here.

Reviewer 2 Report

Review of “Stable isotope analysis of tree rings: a novel simplified protocol for milling wood samples and reducing material loss” by Osvaldo Pericolo, Camilla Avanzi, Andrea Piotti, Francesco Ripullone and Paola Nola [Forests-2150167]

This paper is a technical note specifically related to increasing the throughflow of oak tree-ring wood samples for isotope analysis by grinding more samples together in batches of smaller milling capsules. The paper is really about increasing rate of sample milling. The strengths of the paper are in the rigorous measurement of particle size and the statistical analysis. The many weaknesses of the paper (content and writing) are described below, but they should be resolvable with some significant time and effort on the part of the authors. One potential weakness is that the paper does not show the isotope results from the milled samples in their test of the standard and new protocols, which would have been particularly interesting and informative. However, that might take too much additional work unless the authors already have those numbers in hand.

The authors justify this investigation and potential benefits of this milling advance by means of its application to future studies on water-use efficiency related to “declining trees” associated with drought and future climate change.

1. “declining” and “declining trees” (also “non-declining”) wording is used liberally throughout the paper, but what is it that the authors mean by these terms. I think they should either change the word to something else, or clearly define “declining” in its first use at the beginning of the paper and then indicate the word ‘declining’ will be used to represent that definition throughout the paper. For example, perhaps the authors are referring to declining forests in the sense that the rate of growth of trees in the forest is declining, or the mortality of individual trees in the forest is increasing, or the density of trees in forest stands is decreasing, or the overall health of trees is decreasing, or some combination of these or some other condition(s). Furthermore, the authors do not explain whether the trees are ‘declining’ because of climate change or population-related pressures (e.g., deforestation, wildfire). Are all trees/forests around the world “declining”, or are some of them just changing [location/species composition]? The authors should seriously consider whether they wish to employ the concept of declining trees/forests throughout the paper as justification for the new grinding protocol, when I don’t think it is even necessary.

The “new” protocol for milling featured in the paper is compared to outcomes from a “standard procedure”. The grinding outcome of the new and standard protocols were not found statistically different, and the original ring size/width is the primary factor that influences the final size distribution of both protocols.
2. More context is needed with respect to the extent that the ‘standard’ grinding protocol is widely used in tree-ring isotope studies? Perhaps within the authors’ laboratory it is a standard protocol, but it is not clear to me how widely used the ‘standard’ protocol is outside of their laboratory. Do the authors know of other labs that use the standard protocol? If so, how many? If the standard protocol is rarely used in plant sciences, then this paper is not widely valuable. Perhaps any use of ball mills for grinding in labs around the world, could benefit from this technical note. Could the authors make such a case in the paper so the readers might infer that findings reported in the scientific note are more universal?  Do references 21 and 24 provide any guidance with respect to answers to these questions and context? Perhaps authors should do a bibliographic search to identify the various methodologies currently used for milling/grinding tree-ring samples for isotope analysis, with which they might better be able to judge the impact of the new protocol.

3. In the Abstract the authors state that in “declining trees or trees subjected to extreme events, pointer years are characterized by a scarce amount of wood material”, but ‘pointer years’ can refer to parameters other than ring width, such as missing rings and reaction wood (https://www.wsl.ch/dendro/products/dendro_glossary/Details_EN?id=224&language=English). Furthermore, considering ring size, both particularly large and particularly small rings could be pointer years.

4. line 70, “the minimum weight of wood [up to ten, 9, 20]”, what does “up to ten” mean? “references 110”? “up to 10 years”? “up to 10 trees? “up to 10 mg”?

5. on line 91, by “a 2.0 ml Eppendorf tube, reporting the ID of the wood sample”, do the authors mean “a 2.0 ml Eppendorf tube labeled with the ID of the wood sample”?

6. on line 98, instead of “stove”, the authors might mean “oven” (which is a common piece of equipment found in laboratories, where stoves are uncommonly found in laboratories)

7. on line 105, in the wording “coupled with a stainless steel bead of 300 mm in diameter”, I think the authors mean “30 mm” (300 mm would be a huge bead!)

8. on line 106 in the “Standard protocol” subsection, the authors state “The milling duration was 20 seconds (one cycle)” but on line 138 the milling duration for the new protocol is 60 seconds. This means the standard protocol takes less time for the actual milling step. Furthermore, in Table 1 the grinding time per sample is 10 seconds for standard and 5 seconds for new.  How do authors reconcile these apparent differences in stated duration of the milling step: variously stated as 20, 60, 10, and 5 seconds?

9. lines 163-164 state “particle size at 1% (Dv0.1, minimum diameter), 50% (Dv50, median 163 diameter), and 100% (Dv100, maximum diameter)”, but it looks like “Dv0.1” should be changed to “Dv1” based on the number after “Dv” referring to the percent, i.e. Dv50=50% and Dv100=100%. “Dv0.1” would be 0.1% based on the scheme of these designations.

10. on lines 208-209, the measurement of size range of particles produced by standard and new milling protocols is an interesting and important parameter the authors have determined, and it ranges from about 1.3 µm to 3500 µm. This is quite uneven and the 3500 µm is 3.5 mm, which is huge by most wood isotopes grinding standards. This paper is about milling, but what does this large size range mean with respect to the next step of isotope analysis, namely either the isotope measurement on whole wood tissue or the preparation of cellulose before the isotope measurement? What does a plotted distribution of these sizes look like? The paper only gives minimum, max, mean and median.

11. on line 225 and elsewhere, the “χ2” does not look right. It looks like a standard-size ‘2’. Shouldn’t the ‘2’ be a superscript, like χ2 instead of χ2?

12. on lines 245-246, the sentence “Despite an efficient method to improve isotopic analysis related to cellulose extraction has been recently proposed [24], literature and guidelines on…” would be better written as “Despite a recently proposed [24] efficient method to improve isotopic analysis related to cellulose extraction, literature and guidelines on…”

13. lines 257-258, the wording “when there is a little amount of wood material available,” would be better as “when the amount of wood material available is very small,”

14. on lines 273-277, the authors state that high inter-tree isotope variability has been observed and that somehow the ability to analyze more trees because of the new protocol described in this paper and “help clarify the interpretation of such variability”. It is not clear to me how this would work, and I think the authors are grasping for justification. In my mind, much of the inter-tree isotope variability is related to microenvironments, architecture, health, and genetics of the individual trees. If this is actually true, I don’t understand how the new protocol would clarify this. Perhaps the authors just mean that because of the possibility of more trees being analyzed using the new grinding protocol, the population variance may be better defined(???), which would be a worthwhile justification.

15. on line 288-289, the wording “when there is little amount of wood material available,” would be better as “when the amount of wood material available is very small,”

16. on lines 296-300, again the authors seem to be grasping for examples of benefits of the new protocol. The authors indicate evolutionary research “would greatly benefit from adding water use efficiency measurements to the battery of dendrophenotypic traits investigated.”  Doesn’t ecological and evolutionary research already use water-use efficiency measurements??

17. I think the authors need to rethink the two paragraphs of the Conclusions and present their conclusions in terms of their actual findings, rather than marginal relevance to hypothetical applications that are not the subject of this paper, which currently dominates their Conclusions.

REFERENCES

18. journal titles show many different formats, some are abbreviated, some have titles in which all words begin with upper-case letters, and some have upper-case letter in the first word of the title but not in the others. Authors will need to settle on uniform and consistent formats for journal titles.

19. Formatting of journal article titles and chapter titles is not consistent; some have only the first letter of the first word as upper case, and others have most of the words in the title beginning with upper-case letters.

20. Latin names should be italicized on lines 335 and 349.

Round 2

Reviewer 1 Report

Your responses to the comments:

#1.13 I do not think that keywords should be chosen in that way. Your article is not about phenotypic characterization, so remove this keyword from the list.

#1.30 You at least have to mention in your manuscript that it is necessary to further check if the protocol has an impact on the isotope values of the pre-processed samples. Until now, it is not clear and you cannot exclude any impact.

#1.52 I actually think that the jitter contains lots of information - not provided by the boxplot (per definition). You can include jitter points without filling, for example, or in grey instead of black. As visible from this image, there are a lot of outliers (not only seemingly one as depicted from the boxplot itself). Furthermore, the boxes do not show the distribution of points as visible in Fig. d III/IV - there are mainly data points on the extremes of the box, resulting in a large (but mainly empty) box. I strongly advise to include the jitter and discuss the outliers and the distribution of the data detected there.

Why is there something like a limit on a maximum particle size of 3500 µm, with several points falling exactly at this size? 1) Why exactly 3500 µm? Is it a measurement bias (measurement accuracy etc.)? You also need to discuss this in your manuscript.

In addition, I would like to point to Fig. a, size class IV, new protocol: The jitter dos not match the boxplot (see the position of the median of the boxplot and where your points are - something went wrong, check all boxplots)

#1.63/1.65 The risk of contamination due to cleaning the milling balls is then given for both protocols. This needs to be mentioned in the text.

#1.67 I do not think that a mean loss of >50 % (up to 80 %) is realistic or what is generally observed with the old protocol, i.e., as you mention a large amount of time for removing sample material from the milling jars/balls. I do not know how this level of loss can result (besides insufficient attempts to remove sample material at the walls of the jar etc.). From my experience, I can say that sample losses from milling should be much lower than 50-80 %.

Additional comments to the revised manuscript:

l. 23 “taken from living trees” - not sure if this is understandable to the readers

l. 44-56 Your article is not at all about isotope analysis. There is not reason why you include information on δ13C, δ18O, or δD in the introduction.

Table 1 The time is still very confusing and it is impossible to really track how you measured/approximated it. Please think about another way to illustrate the required time per step.

Table 1 1s for washing the small steel beads of the new protocol? Why is this time so much lower than that of the old protocol (49.5 s - where does this time result from?)? You have to manually clean the beads first, put them in the jars, have a milling run with ethanol - that is the same for both protocols, I guess. Where is the difference? Furthermore, be aware that insufficient cleaning of the small beads may result in high levels of contamination.

Figure 2 As mentioned above, include the jitter and discuss the many outliers as well as the maximum size of 3500 µm.

Figure 3 Thank you for highlighting this. What is not clear to me from this figure: Are these the (combined) Raman spectra of n = 10 Eppendorf tubes, n = 10 old protocol measures, and n = 10 new protocol measures? Please indicate this in the Figure caption.

Reviewer 2 Report

The paper has been substantially revised and improved by clarification of wording. However, as I read through the revised paper, I began to think I do not understand how the authors were distinguishing ‘pretreatment’, ‘pre-processing’ and ‘milling’ procedures in the scheme of this paper.

 1. The title now contains “pre-processing” and “milling”. What is the difference?  If pre-processing is the polishing and cutting of the rings, isn’t pre-processing exactly the same in the standard method as the new method?  It is the ‘milling’ procedure that differs between the standard and new methods. Thus title should state the paper is presenting a new protocol for ‘milling’ only. However, on line 26, I get the idea that authors are even using ‘pre-processing’ for something else, i.e., all the steps prior to any chemical processing to extract cellulose.  What do authors mean by “pre-processing” and does it mean the same thing throughout the manuscript? On top of that, what do authors mean by “pretreatment” on line 85 and how it is related to the other terms, pre-processing and milling.

2. line 18 should read “by extremely narrow rings”

3. line 24, “was freezing” rather than “were freezing”

4. on further reflection on the wording on lines 112-113, “inside a grinding jar, coupled with a stainless steel bead of 30 mm in diameter”, that 30 mm is 3 cm, which is quite large. In the Fisher Scientific catalog I see they sell balls that are 2.5 cm in diameter (25 mm), https://www.fishersci.com/shop/products/mm-400-mixer-mills-grinding-balls/0841789

Maybe it would be worth verifying the 30 mm size stated in the paper.  Also, at a size of either 2.5 cm or 3 cm, I would not call it a “bead”…it is a full-fledged grinding ball.

5. line 120, change wording to “several variants of such a procedure, but our aim here was to compare” [by adding the word “but”]

6. line 121-122, in their response letter, the authors did not provide any evidence that the standard protocol used in this comparison is “a protocol that includes the most frequent features described in the literature”. Given that they have not provided any numerical evidence, I suggest the wording be simplified to “a frequently used ball-milling protocol described in the literature”

7. line 128, “they were moved into” or “they were inserted into”?

8. Lines 134-135, “Quickly milling the samples was required to minimize warming after removal from the freezer.”

9. line 136, “reason the time at room temperature is prolonged, it is”

10. line 165, “Panel A shows the Mixer Mill MM 400”

11. lines 209-210, “Freezing time was not considered because it is not counted as”   [add ‘it is’]

12. lines 258-259, by “optical approach”, do authors mean they checked for foreign particles by “microscopic examination”?

13. line 263, “Certainly, the short milling time and”

14. lines 292-295 are a bit awkward, and I am not sure I understand the point. Perhaps if the authors wish to try to retain this justification, it could be better worded as “and, therefore, the full growing season will be better represented.”

15. line 296, do authors mean “interrupted by temporary summer dormancy”
